# High-Flow Nasal Cannula System in Respiratory Failure Associated with Interstitial Lung Diseases: A Systematic Review and Narrative Synthesis

**DOI:** 10.3390/jcm13102956

**Published:** 2024-05-17

**Authors:** Raffaella Pagliaro, Luigi Aronne, Ramona Fomez, Vincenzo Ferri, Antonia Montella, Stefano Sanduzzi Zamparelli, Andrea Bianco, Fabio Perrotta

**Affiliations:** 1Department of Translational Medical Sciences, University of Campania “L. Vanvitelli”, 80131 Naples, Italy; raffella.pagliaro@studenti.unicampania.it (R.P.); ramona.fomez@studenti.unicampania.it (R.F.); vincenzo.ferri@studenti.unicampania.it (V.F.); antonia.montella@studenti.unicampania.it (A.M.); andrea.bianco@unicampania.it (A.B.); 2U.O.C. Clinica Pneumologica L. Vanvitelli, A. O. dei Colli, Monaldi Hospital, 80131 Naples, Italy; 3Division of Pneumology, A. Cardarelli Hospital, 80131 Naples, Italy; stefano.sanduzzizamparelli@aocardarelli.it

**Keywords:** interstitial lung diseases, high-flow nasal cannula, acute respiratory failure, palliative care, mortality

## Abstract

**Background**: High-flow nasal cannula (HFNC) therapy has emerged as a promising treatment modality for interstitial lung disease (ILD)-related respiratory failure. This systematic review aims to evaluate the efficacy and safety of HFNC therapy in patients with ILDs. **Methods:** A comprehensive literature search was conducted using major electronic databases to identify relevant studies investigating the use of HFNC therapy in ILD patients with respiratory failure. Outcome measures of interest included improvements in oxygenation, dyspnea relief, respiratory rate control, hospital length of stay, and mortality. **Results:** Twelve studies were analyzed with an overall population of 715 patients included. Idiopathic Pulmonary Fibrosis (IPF) was the most prevalent type of ILD. Evaluated clinical settings were acute (7 studies), chronic (2 studies), and end-stage (3 studies) ILDs. The HFNC as a support for acute respiratory failure seems not inferior to non-invasive ventilation while offering better comfort and patient’s perception. Poor data are available about use in chronic/long-term or rehabilitative settings. In end of life/palliative care, an HFNC might improve quality of life. Despite the promising results, further research is warranted to establish optimal HFNC protocols, identify patient subgroups most likely to benefit, and explore long-term outcomes. **Conclusions:** Overall, the HFNC appears to be a valuable therapeutic option for managing respiratory failure in ILD patients, offering potential improvements in oxygenation and symptom relief.

## 1. Introduction

Interstitial lung diseases (ILDs) are a group of diffuse parenchymal lung conditions characterized by alveolar type II exhaustion, inflammation, and scarring of the lung tissue, leading to progressive respiratory dysfunction and respiratory failure [1]. Patients with ILDs often experience acute exacerbations of their condition, necessitating prompt and effective management strategies to improve outcomes and alleviate symptoms [2]. In ILD patients, an impairment in gas exchange is a common finding, reflecting an increased alveolar arterial oxygen gradient [3]. Hypoxemia in ILDs consists of multiple physiologic derangements including diffusion impairment, mismatches between ventilation and perfusion, and abnormalities of the pulmonary vasculature leading to pulmonary hypertension [4]. Chronic repetitive inflammatory processes and abnormal tissue healing can cause gradual damage to the alveolar units, resulting in fibrosis and reducing the ability of oxygen to pass from the alveoli to the small blood vessels. Pulmonary function tests are mainly characterized by a restrictive pattern with decreased forced vital capacity (FVC) and total lung capacity (TLC) associated with reduced diffusion lung capacity for carbon monoxide (DLCO). These lung alterations cause an increase in respiratory rate (RR) as a compensatory mechanism; the lung compliance reduction, due to the related increase in pulmonary elastic return, also contributes to the RR elevation [5]. HFNC therapy has emerged as a potential therapeutic option for respiratory failure associated with ILDs [6]. Consequently, providing supplementary oxygen often becomes a primary treatment strategy for patients with both acute and chronic respiratory failure [7,8]. This review aims to evaluate the current evidence in selected settings of patients with ILD treated with HFNC.

### 1.1. Acute, Chronic, or Acute on Chronic Respiratory Failure in Patients with Interstitial Lung Diseases

The natural history of ILDs is characterized by a variable and often progressive course with symptoms such as worsening of dyspnea, dry cough, and fatigue. Acute respiratory failure (ARF) is a potential manifestation of an ILD both in the context of acute onset ILD and/or acute exacerbations (AEs) of a chronic fibrosing ILD. An AE has been defined as an acute, clinically significant respiratory impairment of unidentifiable cause. While AEs may develop in several fibrosing ILDs, most of current knowledge is related to IPF-AEs and diagnostic and therapeutic management recommendations are transposed to other fibrotic ILDs because of the lack of specific literature. In 2007, Collard et al. proposed a diagnostic criteria of IPF-AE encompassing a significant clinical worsening within 30 days associated with a novel radiologic abnormality on a high-resolution computed tomography (CT) such as bilateral ground-glass opacification or consolidation, excluding alternative etiologies like infection, heart failure, and pulmonary embolism [9]. In IPF, several risk factors have been documented including low FVC, low DLCO, low 6 min walk distance, pulmonary hypertension, elevated serum level of Krebs von Lungen-6 (KL-6), and also a prior history of acute exacerbation [10,11,12]. In particular, in patients with ARF and ILD three possible scenarios have to be distinguished: ARF in known chronic ILDs, unknown chronic ILDs presenting with ARF, and de novo acute ILDs presenting with ARF. In clinical practice, a diagnostic work-up based on laboratory tests, a CT scan (with an angiography component if pulmonary embolism is suspected), and bronchoscopy for microbiological cultures or lavage are generally considered. This approach helps to assess all potential causes contributing to the respiratory distress [13]. Song et al. showed that rapid deterioration (RD) is relatively common during the course of IPF and acute exacerbation (AE) is the most frequent cause of RD, followed by infections [14]. Therefore, oxygen supplementation is the mainstay of treatment of ARF in ILDs. Notably, ILDs may also be complicated by chronic respiratory failure (CRF), especially in the advanced stages. CRF in ILD patients may reflect the extension of the ILD and mechanic consequences (elevated pulmonary elastance and increased alveolar arterial oxygen gradient) as well the burden of comorbidities such as pulmonary hypertension, chronic obstructive pulmonary disease (COPD), heart failure, and lung cancer. Some research on CRF-ILD patients has documented that long-term oxygen supplementation (LTOT) improves patient-reported outcomes (PROs), quality of life (QoL), and exercise tolerance [15,16]. Most recently, HFNC therapy has been used in different settings of patients with ILD; therefore, in this systematic review we decided to evaluate acute conditions, including acute exacerbations, chronic long-term and rehabilitative settings, and end-stage scenarios for palliative care.

### 1.2. High-Flow Nasal Cannula Therapy: Principles and Pathophysiological Advantages

The HFNC has recently emerged as a crucial therapeutic support strategy for hypoxemic patients. The HFNC device is an open circuit system, composed of a flow generator (up to 60 L/min), an ambient air/oxygen mixer (which allows the delivery of inspiratory oxygen fractions (FiO_2_) up to 100%), an active heated humidifier and gas heater (from 31 to 37 °C), and a heated disposable circuit connected to a nasal cannula [17]. The main advantages of HFNC therapy (Table 1) are:Effective delivery of the selected FiO_2_. High flows, by matching or exceeding the patient’s peak inspiratory flow (PIF), prevent the administered gas mixture from being diluted by ambient air [18]. This ensures that the fraction of inspired oxygen is equal to the one provided.Reduction in the resistive respiratory work required to overcome the elastance of the nasopharynx during inspiration. Administering the gas mixture at a flow rate equal to or greater than the patient’s peak inspiratory flow (PIF) allows the counteraction of the inspiratory resistance resulting from the tendency of the nasopharyngeal walls to retract [19].Creation of a positive pressure in the airways that increases linearly with the administered flow and is significantly higher when the subject breathes with a closed mouth [20].Continuous removal of CO2 from the nasopharyngeal dead space (wash-out effect) is facilitated using nasal cannulas that do not occlude more than 75% of the nostrils, which allow the escape of gases subject to “washing out” [21]. Consequently, the patient can always inhale ‘fresh’ gas, avoiding the rebreathing of previously exhaled CO_2_ residing in the dead space, thereby reducing the inspiratory fraction of CO_2_ (FiCO_2_) and increasing that of oxygen (FiO_2_) [22]. Moreover, the wash-out of dead space increases the portion of minute volume participating in alveolar ventilation, thus increasing the efficiency of respiratory effort [19]. This leads to a reduction in pCO_2_ levels [23].Administration of heated and humidified gas mixtures which, at 37 °C, have a relative humidity of 100%. This counteracts the harmful effects of low-temperature gases (harm to the respiratory mucosa, increase in pulmonary vascular resistance, and increase in reactivity of the bronchial muscles) and anhydrous ones (dehydration and thickening of secretions and a reduction in mucociliary clearance). The heating and humidification of the gaseous mixtures ensure a reduction in the metabolic work required for gas conditioning, fluidification of secretions, and improvement of the mucociliary clearance [18].Non-interference with oral hydration and nutrition of patients.

These advantages, in combination with enhanced patient comfort in comparison to non-invasive or invasive ventilation methods, suggests that an HFNC could be considered as an optimal respiratory support option for patients experiencing decompensated hypoxemic respiratory failure due to an ILD. In conclusion, an HFNC can help to prevent the deterioration of lung function and endotracheal intubation because of its clinical (such as the best comfort for the patient) and physiological benefits (such as alveolar recruitment, humidification and heating, increased secretion clearance, and a reduction in dead space) [24].

## 2. Material and Methods

### 2.1. Eligibility Criteria

This systematic review followed the Preferred Reporting Items for Systematic Review (PRISMA) guidelines (Figure 1) [25]. We applied the following inclusion criteria: experimental studies (randomized, prospective, and retrospective trials) that examine the treatment of HFNC in patients with ILDs; adult (aged ≥ 18 years). The outcome measures included improvements in oxygenation and ventilation at time defined by the study, mortality, modified Medical Research Council (mMRC) score, Borg score, health- related quality of life (HR-Qol), quality of dying and death (QODD), endurance time (CWRET), and distance covered during the six-minute walk test (6MWT) after HFNC therapy. We therefore present a narrative synthesis of the systematic review. Study registration was waived due to the narrative description of the collected research.

### 2.2. Study Selection and Quality Assessment

Two authors (RP and RF) independently performed article selection by title and abstract screening based on predetermined eligibility criteria. The references of the included studies were manually reviewed for additional eligible studies. Disagreements relating to any aspect of the data extraction process were discussed and resolved by a third reviewer (VF), with the final decision made by consensus. The full-text articles of the selected studies were reviewed independently for the final study selection. The data were extracted and analyzed from the included studies (RP, RF, and VF). The quality of the studies included was assessed using the Newcastle–Ottawa Quality Assessment Scale for cohort studies or adapted for cross-sectional ones. This scale consists of three items: selection, comparability, and outcome. According to the different criteria, a maximum number of stars can be attributed for each item with a maximum total number of 9 stars. Studies with a total score ≥ 6 were considered as high-quality studies while scores less than 4 categorized studies as unsatisfactory.

## 3. Results

### 3.1. Study Characteristics

Study characteristics are described in Table 2. We included twelve studies regarding HFNC usage in various setting of ILDs: end-stage/palliative care (three studies), chronic respiratory failure (two studies), and acute ILD-related respiratory failure (seven studies). Between these studies, only one evaluated patients both in acute and end-stage settings, using a 30-day survival rate after the beginning of HFNC treatment or Non-Invasive Positive Pressure Ventilation (NPPV) as the primary endpoint. Seven studies were retrospective (one of these was multicenter). A total of 715 patients (70.3% males) were included. IPF was the most prevalent cause of respiratory failure among the different ILDs examined in this systematic review. Seven studies compared oxygenation parameters in patients treated with non-invasive ventilation (NIV) vs. those treated with an HFNC (Table 3).

### 3.2. High-Flow Nasal Cannula in ILD Patients with Acute Respiratory Failure

Acute exacerbations (AEs) of ILDs are the most common causes of respiratory deterioration. In order to improve oxygenation and ventilation in patients with ILDs and ARF, different studies have evaluated the benefits of non-invasive respiratory supports. However, the role of an HFNC in ARF due to ILDs is not fully explored. Firstly, patients’ comfort and tolerance of the respiratory support are necessary for successful respiratory management. The introduction of an HFNC as therapy in AE-ILD patients showed some advantages such as minor discontinuation of therapy, less use of analgesics and sedatives (31.6% vs. 78.6% *p* = 0.001), and better oral intake (23.3% vs. 52.8%, *p* = 0.003) when compared to mechanical ventilation (MV) [35]. In addition, the use of MV with positive end-expiratory pressure (PEEP) values > 10 cm H_2_O was associated with a higher mortality in ILD patients with acute exacerbations [38]. Koyauchi et al. evaluated the efficacy of an HFNC in a retrospective study analyzing a cohort of 66 patients with AE-ILDs. Authors reported an improvement in oxygenation with a subsequent withdrawal of the HFNC in 26 patients (39.4%). Less long-term oxygen therapy (LTOT) before AE-ILD (*p* = 0.045) as well as longer hospital stays (*p* < 0.001) were associated with a higher probability of HFNC success. Interestingly, the SpO_2_/FIO_2_ ratio of ≥170.9 after 24 h of initiating HFNC treatment emerged as a promising predictor of HFNC success (30-day survival rate: 70.3% vs. 20.7%, *p* < 0.001) [31]. According to tolerability, an HFNC may represent a more tolerable option for patients with ARF due to AE-ILDs. This was demonstrated in a cohort of 84 patients admitted for hypoxemic ARF due to an ILD. Specifically, compared to NPPV, HFNC treatment was correlated with significant lower interruption and discontinuation rates (3.7 vs. 23.3%; *p* = 0.009 and 0 vs. 10%; *p* = 0.043, respectively), despite an equivalent survival rate amongst the two groups. Moreover, it should be considered that the possibility of eating and having conversations for patients treated with HFNC is different from NPPV patients [26]. These results were confirmed by Imai et al. who evaluated patients with ILDs in two device settings: HFNC vs. NPPV; indeed, patients treated with NPPV tended to have a longer hospital stay compared with those treated with an HFNC. In addition to this, more patients receiving an HFNC remained on oral intake [14 (67%) vs. 2 (14%) patients, *p* = 0.002] and did not have cognitive dysfunction or coma [14 (67%) vs. 4 (29%) patients, *p* = 0.03] within the last 24 h before death than patients receiving NPPV [36]. Two different studies investigated the role of an HFNC in an ICU setting. Vianello and colleagues failed to find a significant difference in terms of survival between patients receiving conventional oxygen therapy vs. HFNC therapy (median survival time: 133.0 (95% CI, 26.0–374.0) vs. 21.0 (95% CI, 13.0–61.0) days; *p* = 0.1323) in a small cohort of 17 patients with ARF due to AE-IPF. Attempting to analyze the possible causes of HFNC failure, authors found that high CRP levels above 100 mcg/mL at the time of respiratory ICU admission posed a substantially higher risk of HFNC failure. However, the study proved that an HFNC can be associated with a short-term mortality lower than 50% in the event of ARF, especially in patients who are not responding to conventional oxygen therapy [32]. Similar results were reported by Ji-Hoon Lee et al. The authors compared in-hospital mortality during HFNC therapy and MV in a cohort of 61 IPF patients with ARF. The mortality rate was 53.3% for the HFNC group and 55.6% for the MV group, without a significant difference. However, study showed a significantly shorter length of hospitalization and ICU stay (*p* > 0.001) amongst patients treated with an HFNC (13 vs. 24 *p* = 0.134) [33]. Conversely, a small single-center retrospective observational study has suggested potential survival benefits in a cohort of 32 patients with ARF secondary to ILDs, such as IPF, nonspecific interstitial pneumonia (NSIP), Connective Tissue Disease (CTD-ILDs), and Fibrotic Hypersensitivity Pneumonia (F-HP) treated with an HFNC (n = 13) compared to NPPV (n = 19). Although both length of hospitalization [8 days vs. 7 days (*p* = 0.81)] and 30-day intubation rate [8% vs. 37%: (*p* = 0.069)] were reported to be not significantly different between the NPPV group in comparison with HFNC patients, respectively, 30-day mortality was reported as 23% and 63% in patients treated with an HFNC vs. those receiving NPPV, respectively (*p* = 0.026) [34]. Data interpretation on mortality should be read with caution based on study design and the small number of studies; furthermore, in this study, the HFNC was titrated to a median gas flow rate of 50 L per minute with a median FIO_2_ of 0.45 (interquartile range, 0.40–0.63) and no data about humidity of the flow have been reported [34]. Finally, Shebl et al. examined the rate of intubation in patients in therapy with an HFNC vs. NIV; in particular, the authors identified that this rate was 20.6% (7 out of 34 patients) in the HFNC group and 22.2% (8 out of 36) in the NIV group (*p* = 0.87). As secondary outcomes, they found that the ventilator-free days at day 28 was higher in the HFNC group (20 ± 5 vs. 16 ± 7 days in the NIV group; *p* = 0.008) and the rate of in-hospital mortality was 26.5% in the HFNC group vs. 30.6% in the NIV group. In this study, an HFNC did not result in a decrease in the rate of intubation among patients with ILDs during the ARF episode but an HFNC allowed more ventilator-free days among those patients when compared with NIV [37].

### 3.3. High-Flow Nasal Cannula in the Management of Chronic Respiratory Failure Secondary to Fibrosing ILD

Supplemental oxygen therapy has traditionally been employed to manage dyspnea and enhance exercise tolerance in patients with ILD [39]; emerging research has explored the potential role of HFNC chronic therapy as an alternative or additive treatment modality [40,41]. In a small study, Weinreich et al. assessed the impact of HFNC—titrated to at least 30 L/min with a humidity temperature of 37 °C—on walking distance, dyspnea sensation, lung function, and blood gas analyses. They evaluated 10 patients with ILDs for a period of 6 weeks and with a use of at least 6.5 h per day. In particular, HFNC therapy demonstrated significant improvements in exercise performance and dyspnea sensation, highlighting its potential efficacy in managing ILD-related symptoms. Additionally, the study observed favorable changes in the mMRC score, Borg score, and distance covered during the 6MWT after HFNC therapy, indicating an overall enhancement in exercise capacity and respiratory comfort. However, no significant improvement in lung function was found [29]. Another study evaluated the impact of HFNC on exercise capacity in fibrotic ILD patients. The authors performed a prospective randomized controlled cross over trial with a constant work rate endurance test (CWRET) and enrolled 20 patients divided into HFNC groups and Venturi mask (VM) groups. Despite the expectations, the study did not achieve the predefined endpoints. Nonetheless, many patients responded positively to the HFNC, with superior effects observed in some cases compared to the VM. Subgroup analysis revealed that the HFNC significantly prolonged endurance time compared to the VM. Notably, the HFNC enhanced endurance time and exertional dyspnea in some patients. Potential mechanisms for these benefits include the clearance of physiological dead space, leading to improved respiratory function. These findings underscore the importance of supplemental oxygen in enhancing exercise capacity in fibrotic ILD patients. In conclusion, while the HFNC did not surpass the VM in efficacy in this study, it holds promise as a complementary intervention [30].

### 3.4. High-Flow Nasal Cannula as Support in Exercise Testing and Pulmonary Rehabilitation for Patients with ILD

Pulmonary rehabilitation intervention is part of a multimodal and comprehensive treatment for tackling dyspnea and physical limitations in advanced ILD patients. The added value of an HFNC in supporting muscular and respiratory exercise is not extensively defined as few studies have been published. The first original research in fibrotic ILDs—study population including IPF (60%), iNSIP (5%), CTD-ILD (10%), and unclassifiable ILDs (25%)—found no specific advantages in the overall population when supported with an HFNC or COT despite authors finding a subgroup of patients having an higher benefit from an HFNC. Subsequently, in a small, single-center, cross-over prospective study,10 patients with IPF underwent submaximal CPET (75% of their maximum exercise capacity) on two consecutive days with different oxygen supplementation (HFNC vs. COT). Authors found a significant improvement in the endurance time and an increased inspiratory capacity when patients were supported with an HFNC (*p* = 0.013). These data were recently corroborated in a larger trial in IPF patients, reporting a significant impact on endurance time and SpO_2_ when IPF patients were supported with an HFNC. This provided the rationale for using an HFNC during pulmonary rehabilitation [42,43]. Nevertheless, no study showing the efficacy of HFNC use during rehabilitative intervention for patients with interstitial lung diseases has actually been published or included in the present systematic review.

### 3.5. High-Flow Nasal Cannula in End of Life and Palliative Care for ILD Patients

End-stage ILD refers to the advanced phase of interstitial lung disease; at this stage, lung function is significantly impaired. Management at this stage often focuses on palliative care to alleviate symptoms and improve quality of life. Breathing comfort is a high end of life priority for these patients; therefore, it is important to choose end of life respiratory modalities that improve the quality of dying and death (QODD). The major options in this setting were evaluated by Koyauchi et al., including COT, HFNC, NIV, and IMV. According to the authors, they evaluated 177 patients with ILDs divided into four groups: COT (n = 62), HFNC (n = 76), NIV (n = 27), and IMV (n = 12); symptoms and quality of life were evaluated with QODD rated by the Good Death Inventory (GDI), a tool for measuring QODD from the perspective of the family. The average score of 18 domains of the GDI for QODD was the highest for the HFNC group (4.58 ± 0.67), followed by the NIV (4.38 ± 0.71), COT (4.09 ± 0.96), and IMV (3.96 ± 0.75) groups. Similarly, the score of the “physical and psychological comfort” domain was the highest for the HFNC group (4.55 ± 1.43), followed by the NIV (4.19 ± 1.63), COT (3.34 ± 1.70), and IMV (3.17 ± 1.47) groups [27]. Another study by Koyauchi et al. was conducted on 84 patients affected by IPF, CHP, Sarcoidosis, and CTD-ILD and divided into two groups: the HFNC group (n = 54) and the NPPV group (n = 30). They demonstrated that the 30-day survival rate was equivalent in the two groups (HFNC 31.5% vs. NPPV 30.0%; *p* = 0.86) but a significant improvement in the respiratory rate was observed after beginning HFNC therapy (pretreatment 28.0 ± 7.0 per min vs. posttreatment 23.8 ± 4.3 per min) compared to the NPPV group (pretreatment 31.2 ± 8.1 per min vs. post-treatment 32.6 ± 8.9 per min) [26].

In the setting of palliative care, other authors compared the efficacy of an HFNC with COT in improving dyspnea of patients who had ARF. Patients were randomized to COT for 60 min, followed by an HFNC for 60 min. The primary outcome was the modified Borg scale score. At 60 min, the mean modified Borg scale score in patients receiving COT and an HFNC was, respectively, 4.9 and 2.9. Also, respiratory rates were lower with the high-flow nasal cannula (mean difference 5.9; 95% confidence interval 3.5 to 8.3), showing the superiority of an HFNC to COT, in terms of reduction of symptoms and, therefore, in better quality of live, even in palliative care during the end-stage [28].

## 4. Discussion

The aim of this systematic review was to evaluate the evidence to date on the impact of an HFNC on selected settings of patients with ILDs. During acute ILD exacerbations, an HFNC should be carefully integrated into management of ARF. An HFNC has been associated with a 50% reduction in short-term mortality in patients who are not responding to conventional oxygen therapy in the event of ARF [32]. The 30-day mortality was 23% and 63% in patients treated with an HFNC vs. those receiving NPPV, respectively (*p* = 0.026) [34]. Compared to NPPV, HFNC therapy is more comfortable and allows patients to perform daily activities like eating, drinking, and talking. Improved tolerance of therapy leads to better short-term outcomes, as previous studies have shown that poor tolerance of ventilatory support is associated with higher mortality rates in patients with ARF [44]. According to the FLORALI study, there was a significant difference in favor of an HFNC in 90-day mortality compared to NPPV [45]. In the case of failure of non-invasive respiratory support, an invasive respiratory support should be considered despite potential complications (i.e., infections, ventilator-induced pneumonia, and barotrauma/volutrauma). In IPF, AEs requiring tracheal intubation and IMV, lower partial pressure of arterial carbon dioxide, higher pH, and a less severe APACHE II score at the time of MV initiation were associated with significant advantages in mortality [46,47]. While at this time, data about an HFNC’s role as part of de-escalation strategy after extubation is limited, the above-reported physiological effects, as well the chance to use an interface for a tracheostomized host, should be considered by clinicians in the trials of respiratory support de-escalation along with NPPV.

An HFNC was linked to improved comfort levels, reduction in the severity of dyspnea, and lower respiratory rate. This improvement is likely due to the heating and humification of the air which prevented thick secretions and atelectasis, and also from the low-level of PEEP, which helps keep the airways open and flushes the upper airway dead space [48]. Many studies reviewed suggested that an HFNC provided superior oxygenation as compared to COT. An HFNC was better tolerated and was associated with increased patient comfort and decreased dyspnea in the studies in our review. The HFNC impact on mortality remained better than other oxygen support.

An HFNC is seen as a modern approach to deliver oxygen therapy, offering advantages over COT in several ways. Humidified and heated air delivered by an HNFC is thought to optimize the mucosal function of the respiratory tract. Another setting was explored in the AmbOx trial: a prospective, open-label study which demonstrated the efficacy of supplemental oxygen in improving exercise capacity and exertional dyspnea in FILD patients [49]. However, the potential of HFNC therapy in improving exercise capacity in FILD patients remains understudied. Both studies underscore the importance of personalized treatment approaches in the setting of patients with FILDs in chronic therapy [29,30]. Indeed, HFNC therapy represents a novel and potentially beneficial intervention for addressing the challenges of dyspnea and reduced exercise capacity in these patients. The research findings presented in this review highlight the potential role of HFNC therapy in improving exercise capacity among patients with FILDs. In this context, high-quality data from large and homogeneous populations with long-term follow-ups and standardized HFNC protocols are needed. Furthermore, research is awaited to understand the cost-effectiveness in chronic use as well the potential integration into pulmonary rehabilitation programs [50]. Despite the current lack of knowledge on ILDs, data from studies in COPD patients support that HFNC therapy offers benefits in term of reduction in exacerbations, decreased length of stay in intensive care, and less use of medications. In particular, the reductions in acute exacerbations of COPD that result from adding an HFNC for persons with COPD on LTOT will produce both health benefits and cost savings [51]. Cost savings occur because the HFNC device costs are more than offset by reductions in costly COPD exacerbations. These findings suggest that HFNC therapy may also provide cost-saving opportunities in the management of ILDs compared to the use of the traditional nasal oxygen cannula, particularly in scenarios where patients experience acute respiratory exacerbations or require intensive care support [52]. Although direct evidence is lacking, the consistent trends observed across respiratory conditions indicate the potential value of HFNC therapy in improving patient outcomes and optimizing healthcare resource utilization in ILDs.

An HFNC has several characteristics that would greatly contribute to maintaining a better QOL for patients with end-stage ILDs. First, it can relieve dyspnea caused by hypoxia since it allows the administration of a high flow and high concentration of oxygen. Second, an HFNC is a more favorable for terminally ill patients because it allows them to maintain daily activities such as eating and talking. Last, an advantage is the cost- effectiveness of the use of HFNC in palliative care settings.

Our systematic review has several limitations; first, the number of studies included was small, especially for the comparison between HFNC therapy vs. invasive or non-invasive mechanical ventilation. Therefore, no robust conclusion can be generated, and more RCTs with larger sample sizes are warranted. Second, summary estimates were limited by different types of ILDs, which may interfere with treatment response and an overall prognosis. Finally, researchers should reduce heterogeneity in HFNC protocols—flow rates and humidity—for limiting possible data misinterpretation.

## 5. Conclusions

In summary, HFNC therapy could offer a promising alternative to traditional oxygen therapy for patients with ILDs who need both high levels of oxygen and a high flow rate of gas to correct hypoxemia and manage breathing difficulties and tachypnea.

## Figures and Tables

**Figure 1 jcm-13-02956-f001:**
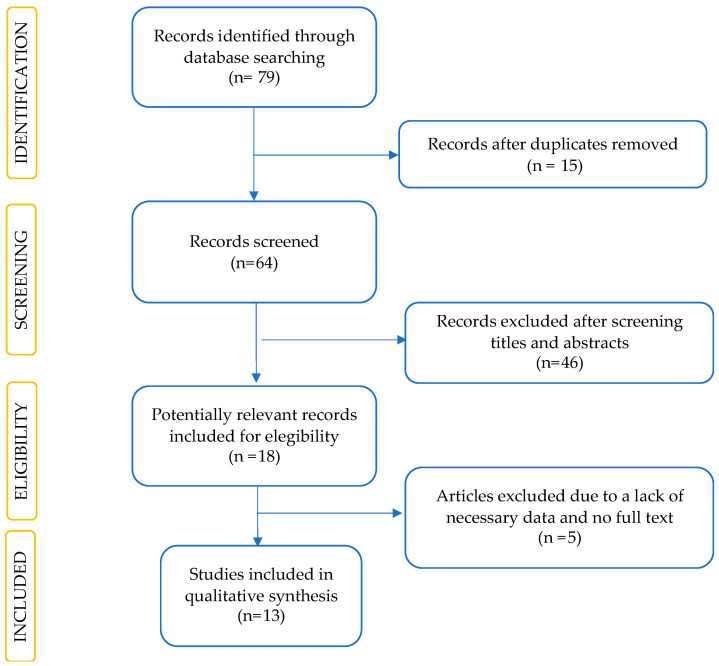
Flow diagram of the article selection procedure based on the PRISMA guideline.

**Table 1 jcm-13-02956-t001:** Advantages and disadvantages of high-flow nasal cannula (HFNC) treatment.

Advantages	Disadvantages
Effective delivery of the selected FiO_2_.	Potential discomfort due to high flow and relatively hot air sensation
Reduction in the resistive respiratory work required to overcome the elastance of the nasopharynx during inspiration	Not immediately available
Provides low positive end expiratory pressure effect	Aerosol-generating procedure that can potentially increase the risk of viral transmission
Carbon dioxide wash-out (reduced anatomical dead space)	
Comfort due to similarity of humidified, warmed air to physiologic conditions of the airway	
Non-interference with oral hydration and nutrition of patients	

**Table 2 jcm-13-02956-t002:** Characteristics of studies included in this review. Not available (NA); U: univariate; M: multivariate; KM: Kaplan–Meier curve; HFNC: high-flow nasal cannula, NIV: non-invasive ventilation; NPPV: non-invasive positive pressure ventilation; COT: conventional oxygen therapy; IMV: invasive mechanical ventilation; VM: Venturi mask.

Authors	Year	Country	Study Design	Number of Patients (Male)	Clinical Setting	Patient Characteristics	Tested Devices	Primary Outcome	Secondary Outcomes	Analysis
Koyauchi et al. (2018) [26]	2010–2017	Japan	Single center; Retrospective	84 (61)	End-stage/ Acute ILD	IPF (44); ILD NO IPF (27); CTD-ILD (10); CHP (2); SARCOIDOSIS (1)	HFNC vs. NPPV	Rates of 30-day survival after be-ginning HFNC or NPPV	Respiratory rate	KM
Koyauchi et al. (2022) [27]	2015–2019	Japan	Multicenter; Prospective	177 (137)	End-stage ILD	IPF (78); ILD NO IPF (58); CTD-ILD (36); CHP (3)	COT; HFNC; NIV; IMV	QODD	GDI	U; M
Ruangsomboon et al. (2019) [28]	2017–2018	Thailand	Single center; Prospective	48 (21)	End-stage ILD	ILD	HFNC vs. COT	Modified Borg scale score	Rating scale score of dyspnea	U
Weinreich et al. (2022) [29]	2019–2021	Denmark	Single center; Prospective	9 (5)	Chronic ILD	FILD	HFNC	Effect of HFNC on patients’ sensation of dyspnea and health-related quality of life (HR-QoL)	6MWT, mMRC	U
Suzuki et al. (2020) [30]	2016–NA	Japan	Single center; Prospective	20 (19)	Chronic ILD	FILD: IPF (17); NSIP (2); CTD-ILD (3); UNCLASSIFIABLE IIP (8)	HFNC vs. VM	Endurance time (CWRET)	Borg scale, HR	U
Koyauchi et al. (2020) [31]	2013_2017	Japan	Multicenter; Retrospective	66 (51)	Acute ILD	IPF (31); non IPF IIP (22); CTD-ILD (11); CHP (2)	HFNC	SpO2/FiO_2_ ratio	30-day survival rate,	U; M; KM
Vianello et al. (2019) [32]	2013–2018	Italy	Single center; Retrospective	17 (14)	Acute ILD	IPF	HFNC vs. NPPV	Mortality Rate	NA	U; M
Lee et al. (2020) [33]	2015–2017	Korea	Single center; Retrospective	61 (48)	Acute ILD	IPF	HFNC vs. NPPV	In-hospital Mortality	NA	NA
Omote et al. (2020) [34]	2011–2017	Japan	Single center; Retrospective	32 (26)	Acute ILD	IPF (18); NSIP (2); CTD-ILD (8); Others (4)	HFNC vs. NPPV	30-DAY MORTALITY	Intubation, ICU length of stay	U; M
Ito et al. (2019) [35]	2009–2012	Japan	Single center; Retrospective	96 (71)	Acute ILD	IPF (26); NSIP (9); CTD-ILD (19); CHP (7); CPFE (13), Other IIP (22)	HFNC vs. NPPV	In-hospital Mortality Rate	NA	KM
Imai R (2019) [36]	2008–2017	Japan	Single center; Retrospective	35 (25)	Acute ILD	IPF (13) CTD-ILD (11) Others IIP (11)	HFNC vs. NPPV	In-Hospital Mortality Rate	Length of hospital stay	NA
Shelbl (2018) [37]	2016–2017	Saudi Arabia	Single center; Prospective	70 (25)	Acute ILD	IPF (19); CTD_ILD (9); CHP (9); Sarcoidosis (7); Other IIP (26)	HFNC vs. NPPV	Need of intubation	In-hospital mortality and ventilator-free days	NA

**Table 3 jcm-13-02956-t003:** Oxygenation parameters and respiratory rate in high-flow nasal cannula vs. non-invasive ventilation.

Author, Year	HFNC PaO_2_ (mmHg) Median (Range) or Mean ± SD	NIV PaO_2_ (mmHg) Median (Range) or Mean ± SD	*p*	HFNC PaO_2_/FiO_2_ (mmHg) Median (Range) or Mean ± SD	NIV PaO_2_/FiO_2_ (mmHg) Median (Range) or Mean ± SD	*p*
Lee et al. (2020) [33]	82.9 ± 47.7	97.6 ± 42.0	0.366	202.9 ± 124.8	196.8 ± 97.3	0.927
Ito et al. (2019) [35]	NA	NA	NA	188 (75–269)	191 (130–313)	0.17
Imai R (2019) [36]	NA	NA	NA	116 (85–163)	160 (82–273)	0.29
Shelbl (2018) [37]				178 ± 55	166 ± 42	0.31
Omote et al. (2020) [34]	62 (56–75)	74 (68–88)	NA	133 (105–158)	144 (114–191)	0.43
Vianello et al. (2019) [32]	69.85 (41.3–258.7)	80.6 (39.0–99.3)	0.831	147 (46–289)	143 (73–248)	0.831
Koyauchi et al. (2018) [26]	NA	NA	NA	100 (79–117)	126 (80–176)	0.25

FiO_2_ = fraction of inspired oxygen; PaO_2_ = partial pressure of arterial blood oxygen; HFNC = high-flow nasal cannula; NIV = non-invasive ventilation.

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
