# Peer review of "High-Flow Nasal Cannula System in Respiratory Failure Associated with Interstitial Lung Diseases: A Systematic Review and Narrative Synthesis"

_jcm, 2024, doi:10.3390/jcm13102956_

Round 1
Reviewer 1 Report
Comments and Suggestions for Authors
This is a review article on the role of high flow nasal cannula system in respiratory failure associated with ILD.
It is a correct review of 13 studies with a correct methodology and results that show the advantages of using high flow HFNC therapies in certain situations of respiratory failure in interstitial pathology, with clear advantages over traditional oxygen therapy and NPPV.
Author Response
Many thanks for your positive evaluation of our manuscript.
Reviewer 2 Report
Comments and Suggestions for Authors
This systematic review meticulously evaluates the efficacy and safety of High Flow Nasal Cannula (HFNC) therapy for patients with interstitial lung diseases (ILDs) experiencing respiratory failure. It highlights the potential of HFNC in reducing mortality rates compared to Non-Invasive Positive Pressure Ventilation (NPPV), especially in cases of idiopathic pulmonary fibrosis (IPF). This paper positions HFNC as a valuable therapeutic option for ILD patients, suggesting its advantages over traditional oxygen therapy and NPPV in improving patient outcomes and quality of life.
However, the review could be improved or expanded upon to enhance its rigor, depth, and clinical applicability:
1. Based on the findings, providing more specific recommendations for HFNC protocols (e.g., flow rates, humidity levels) could guide clinical practice.
2. Highlighting the need for and proposing designs for studies investigating long-term outcomes of HFNC therapy in ILD patients would be valuable.
3. Discussing the economic aspects of HFNC therapy, including comparisons with other forms of respiratory support, could support healthcare decision-making.
Comments on the Quality of English LanguageMinor editing of English language required
Author Response
We thank the referees for their excellent comments. In the revised version we have tried to include their suggestions and in particular
However, the review could be improved or expanded upon to enhance its rigor, depth, and clinical applicability:
- Based on the findings, providing more specific recommendations for HFNC protocols (e.g., flow rates, humidity levels) could guide clinical practice.
Response. We thank the referee for his/her valuable comment. The absence of standardize HFNC protocols is a major issue – we have now included this important consideration in our study limitations. We have tried to include available data about flow rates and humidity/temperature when present into the selected studies.
Once again, many thanks for this very important consideration.
- Highlighting the need for and proposing designs for studies investigating long-term outcomes of HFNC therapy in ILD patients would be valuable.
Response. Many thanks for this comment. We have remodulated the Discussion section integrating the urgent need for robust data in chronic and rehabilitative scenario. We do agree that current literature is biased from heterogeneous group of patients and lack of standardized protocols.
- Discussing the economic aspects of HFNC therapy, including comparisons with other forms of respiratory support, could support healthcare decision-making.
Response. We thank the referee for this valuable comment. This aspect is very intriguing to balance the effects and the costs of a treatment into the management of chronic respiratory conditions. While at this time HFNC Cost effectiveness has been investigated in different scenarios, data from ILDs are lacking. However, In the revised version of the manuscript we have now included data about COPD to offer to the readers the possible advantages of HFNC overt standard treatment also in patients with ILDs. Many thanks for this consideration.
Reviewer 3 Report
Comments and Suggestions for Authors
Respiratory support is an essential therapeutic measure for ILD patients with respiratory failure. his systematic review evaluates the current evidence on the impact of HFNC in various scenarios involving ILD patients, such as acute respiratory failure, chronic respiratory failure, support in exercise testing and pulmonary rehabilitation, and end-of-life and palliative care. The findings indicate that HFNC has been linked to a reduced mortality rate compared to Non-Invasive Positive Pressure Ventilation (NPPV). Moreover, it offers enhanced comfort, enabling patients to engage in their daily activities. This article focuses on a highly important clinical topic, and the findings also contribute to improving the clinical management of ILD combined with respiratory failure. Overall, the manuscript is generally written clearly and concisely. However, there are some minor issues. Minor points
- In the "Discussion" section, the authors should elaborate on how clinicians can judiciously select different respiratory support modalities for ILD with acute respiratory failure, as well as the impact of invasive ventilation on the prognosis of patients with ILD.
- The limitations of this study should be addressed and discussed in the "Discussion" section.
Author Response
We thank the referees for their excellent comments. In the revised version we have tried to include their suggestions and in particular
- In the "Discussion" section, the authors should elaborate on how clinicians can judiciously select different respiratory support modalities for ILD with acute respiratory failure, as well as the impact of invasive ventilation on the prognosis of patients with ILD.
Response. We thank the referee for this valuable comment. We have now included in the Discussion section an overview about HFNC positioning into management of ARF. We highlighted that HFNC has reported in this clinical setting some advantages when compared to COT. We also suggest to integrate HFNC after invasive ventilation as part of de-escalation strategy. We aimed to provide a pragmatic overview which may useful to readers. Many thanks for this very important comment.
- The limitations of this study should be addressed and discussed in the "Discussion" section.
Response. We thank the referee for this valuable comment. In the revised version of the manuscript a discussion of the study limitations has been added. Current literature offers few data – especially in comparison between HFNC therapy and VM – limiting the generation of robust evidence. Also, heterogeneous types of ILDs are generally included and this may lead to potential mislead in data interpretation.